# Research and Clinical Progress of Therapeutic Tumor Vaccines

**DOI:** 10.3390/vaccines13070672

**Published:** 2025-06-23

**Authors:** Chunyan Dong, Zhuang Li, Dejiang Tan, Huimin Sun, Jinghui Liang, Dexian Wei, Yiyang Zheng, Linyu Zhang, Sihan Liu, Yu Zhang, Junzhi Wang, Qing He

**Affiliations:** 1State Key Laboratory of Drug Regulatory Sciences, National Institutes for Food and Drug Control, Beijing 102629, China; dongchunyan@nifdc.org.cn (C.D.); lizhuang0422@163.com (Z.L.); tandj@nifdc.org.cn (D.T.); sunhm@nifdc.org.cn (H.S.); liangjinghui100@163.com (J.L.); 15195222136@163.com (Y.Z.); zly6802024@163.com (L.Z.); zhangyu2024@nifdc.org.cn (Y.Z.); wangjz@nifdc.org.cn (J.W.); 2Department of Experimental Pharmacology and Toxicology, School of Pharmaceutical Sciences, Jilin University, Changchun 130021, China; weidx24@mails.jlu.edu.cn; 3School of Pharmacy, Shenyang Pharmaceutical University, Shenyang 110016, China; 13052799837@163.com

**Keywords:** therapeutic tumor vaccines, clinical progress

## Abstract

Therapeutic cancer vaccines are a new growth point of biomedicine with broad industrial prospects in the post-COVID-19 era. Many large international pharmaceutical companies and emerging biotechnology companies are deploying different tumor therapeutic cancer vaccine projects, focusing on promoting their clinical transformation, and the vaccine industry has strong momentum for development. Such vaccines are also the core engine and pilot site for the development of new vaccine targets, new vectors, new adjuvants, and new technologies, which play a key role in promoting the innovation and development of vaccines. Various therapeutic cancer vaccines, such as viral vector vaccines, bacterial vector vaccines, cell vector vaccines, peptide vaccines, and nucleic acid vaccines, have all been applied in clinical research. With the continuous development of technology, therapeutic cancer vaccines are evolving towards the trends of precise antigens, efficient carriers, diversified adjuvants, and combined applications. For instance, the rapidly advancing mRNA-4157 vaccine is a typical representative that combines personalized antigens with efficient delivery vectors (lipid nanoparticles, LNPs), and it also shows synergistic advantages in melanoma patients treated in combination with immune checkpoint inhibitors. In this article, we will systematically discuss the current research and development status and clinical research progress of various therapeutic cancer vaccines.

## 1. Introduction

Cancer has become a major disease that seriously threatens human health and life worldwide and has always been the core focus of the field of medical research. According to the latest statistics from the International Agency for Research on Cancer (IARC), there were approximately 20 million new cancer cases worldwide in 2022, and nearly 9.7 million people died from cancer [1]. For cancer, traditional treatment methods mainly include surgery, chemotherapy, and radiotherapy. Although they have improved the survival status of patients to a certain extent, their therapeutic effects still have significant limitations. Surgical methods often have difficulty curing metastatic cancers completely. While chemotherapy and radiotherapy kill tumor cells, they also severely damage normal tissues, thereby triggering many adverse reactions and having a significant impact on patients’ lives. In recent years, with the continuous in-depth research on tumor immunology, immunotherapy has gradually emerged and brought new hope for tumor treatment. Such as immune checkpoint inhibitors, cell therapy, tumor vaccines, etc., have become important means of tumor immunotherapy [2,3,4]. Among them, tumor vaccines, as an emerging immunotherapy strategy, have been the focus of much attention in recent years. They specifically recognize and kill tumor cells by activating or enhancing the immune system of cancer patients, thereby achieving the purpose of treating tumors.

Tumor vaccines are those that deliver tumor antigens, such as lysed tumor cells, tumor-associated proteins or peptides, RNA or DNA expressing tumor antigens, into the patient’s body to activate the immune response and exert anti-tumor effects [5,6,7,8]. Tumor vaccines are divided into preventive tumor vaccines and therapeutic tumor vaccines. Preventive tumor vaccines are mainly designed against the pathogens that cause tumors, such as the HPV vaccine for preventing cervical cancer [9,10]. Multiple HPV vaccines have been clinically approved globally. For example, the bivalent vaccine (2vHPV): Cervarix^®^, the quadrivalent vaccine (4vHPV): Gardasil^®^, and the nonavalent vaccine (9vHPV): Gardasil9^®^. Therapeutic cancer vaccines are mainly designed for tumor antigens (tumor-associated antigens, specific antigens), stimulating the body to produce specific immune responses to kill tumors. Such vaccines have the characteristics of broad-spectrum and personalization and have important advantages and significance. For therapeutic cancer vaccines, several marketed products have been launched, such as the BCG vaccine (TheraCys^®^), the dendritic cell vaccine Provenge^®^, the oncolytic herpesvirus vaccine (Imlygic^®^), and the peptide vaccine (Cimavax^®^), which are used to treat prostate cancer, melanoma, and renal cell carcinoma. So far, there are still many therapeutic cancer vaccines under research and development worldwide, mainly including viral vector type, bacterial vector type, cell vector type, peptide type, and nucleic acid type. Various therapeutic cancer vaccines combat tumors by directly killing or enhancing the immune response (Figure 1). For viral vector tumor vaccines, they exert anti-tumor effects through multiple pathways. For example, the direct oncolytic effect is that after the virus invades tumor cells, it replicates in large quantities and eventually lyses the tumor cells [11,12,13]. In addition, viral vector vaccines also inhibit tumors by activating immune responses, altering the tumor microenvironment, and disrupting the blood supply to tumors [14,15,16,17,18,19]. Bacteria also have their unique advantages as delivery carriers. For instance, bacterial vaccines loaded with drugs can spontaneously target and colonize tumor tissues after being absorbed [20]. Vaccines based on bacterial vectors present antigens carried by APCs and are recognized by T cells, which activate CD4^+^ and CD8^+^ T cells, thereby enhancing the immune response and inducing apoptosis of tumor cells [20,21,22,23,24]. The mechanism of action of cell vaccines is to introduce tumor antigens or immune-stimulating molecules into the patient’s body through cell vectors, activate the patient’s own immune system, and induce an immune response to achieve the purpose of controlling or eliminating tumors [25,26,27,28,29]. With the development of molecular biology techniques, the tumor antigens recognized by the immune system have been identified. Designing and developing synthetic peptides corresponding to the antigenic epitopes of tumor-reactive lymphocytes has become an important means of treating tumors. Designing peptide-based vaccines to stimulate anti-tumor T-cell responses has many advantages, such as ease of manufacturing and quality control, as well as showing good safety in existing clinical studies [30]. For effective peptide tumor vaccines, their anti-tumor mechanism of action depends on activated CD8^+^ and CD4^+^ T cells [31,32,33]. Nucleic acid vaccines include two types: DNA vaccines and mRNA vaccines. For DNA vaccines, DNA needs to be transported to the cell nucleus for transcription and then translated into the cytoplasm [34]. Vaccines based on mRNA can be directly expressed in the cytoplasm of transfected cells [35]. Nucleic acid vaccine antigens rely on somatic expression and release and then transfer to local APCs for presentation, activating CD8^+^ and/or CD4^+^ T cells [36]. The main indications of various therapeutic cancer vaccines include prostate cancer, lung cancer, glioblastoma, melanoma, breast cancer, liver cancer, etc. [7,37,38,39,40,41,42]. With the continuous advancement of technology, tumor immunotherapy has developed rapidly and has become a research and development hotspot in the field of tumor treatment. This article will focus on systematically discussing the current research and development status of various therapeutic cancer vaccines and the updated related clinical research progress in the past five years, aiming to provide strategies and new ideas for the research and transformation of tumor therapeutic vaccines.

## 2. The Progress of Therapeutic Cancer Vaccines

Therapeutic cancer vaccines mainly use tumor antigens and immune adjuvants to induce specific immune responses to kill tumor cells, and anti-tumor T cells are the effector cells expected to be induced by such vaccines. Hundreds of therapeutic cancer vaccines are currently under clinical evaluation, including viral or bacterial vector vaccines, cellular vaccines, peptide vaccines, and nucleic acid vaccines. Next, we will systematically discuss the research progress of different types of therapeutic tumor vaccines.

### 2.1. Viral and Bacterial Vectors for Therapeutic Cancer Vaccines

#### 2.1.1. Viral Vector Tumor Vaccines

A viral vector is a tool that uses genetic engineering technology to transform viruses and then infects cells to introduce foreign genes into cells and express genes for a long time. Instrumented viral vectors have been widely used in the field of immunotherapy due to their advantages, such as high transfection efficiency, high expression level of exogenous genes, strong targeting, strong killing effect, and strong immune activation ability [43]. Virus vectors mainly include lentivirus, adenoviruses and adeno-associated viruses, poxvirus, herpesvirus, and oncolytic virus. Viral vector-based therapeutic vaccines for tumors have the following advantages: Due to the natural infection ability of viruses, their antigen delivery capacity is significantly superior to that of non-viral vectors (such as naked DNA, RNA, liposomes, etc.) [18,19]. In addition, their strong inherent adjuvant effect enables the vaccine to strongly activate the body’s innate immune response after immunization, thereby effectively initiating subsequent T-cell-mediated anti-tumor immune responses. The immunogenicity of the vector itself is crucial for breaking through the anti-tumor immune-suppressive microenvironment [20]. However, traditional viral vectors also have certain limitations. Firstly, there is a limit to the antigen capacity of the vector [21]. For very complex antigen combinations (such as dozens or hundreds of personalized neoantigens) or large gene fragments, they may not be fully accommodated. This limits the breadth and complexity of the expressed antigens, especially in the application of personalized neoantigen vaccines [22]. Moreover, humans have been exposed to various common viruses (such as adenovirus and varicella-zoster virus) in daily life, and neutralizing antibodies and/or T-cell immune memory against the components of the viral vector already exist in the body [23]. Pre-existing antibodies will rapidly neutralize the injected vector particles, preventing them from infecting target cells and delivering antigens, resulting in a sharp decline or even ineffectiveness of the vaccine’s immunogenicity. Due to the immune system’s memory of the vector, high-titer neutralizing antibodies will be rapidly produced after the first vaccination, making subsequent booster immunizations with the same vector very difficult or ineffective, which is a core challenge. Currently, there are five marketed oncolytic virus products in the world: Rigvir (Latvia), Oncorine (China), IMLYGIC (USA), Adstiladrin (USA), and DELYTACT (Japan), with indications including melanoma, head and neck squamous cell carcinoma, bladder cancer and glioma [44,45,46,47,48]. Next, we will systematically describe the research progress of therapeutic cancer vaccines based on viral vectors (Table 1).

##### Adenoviruses and Adeno-Associated Viruses

Adenovirus vectors can efficiently deliver tumor-associated antigens (TAAs) or tumor-specific neoantigens, inducing a strong T-cell immune response. For example, Ad5-E1A-based adenovirus vector vaccines have shown good antigen delivery in breast and ovarian cancer. Recurrent respiratory papilloma (RRP) is a stubborn neoplastic disease associated with chronic HPV6 or 11 infection, causing severe hoarseness and airway obstruction, and there is no approved therapy [49]. PRGN-2012 is a new type of gorilla adenovirus immunotherapy drug that can enhance specific T-cell immunity against HPV 6/11 [50]. In the Phase 1 clinical trial (NCT04724980), PRGN-2012 was first used to treat severe and invasive RRP in adults and showed good clinical benefits. It was generally safe, and the complete response rate in the highest-dose group reached 50% [50].

Adenovirus vector vaccines against cancer are a strong area of preclinical and clinical research. There are many studies on therapeutic cancer vaccines based on adenovirus vectors that have entered the clinical stage, but most of them are in the clinical Phase 1–2. Adenovirus vector vaccines are mainly used for the treatment of glioblastoma (NCT05686798, NCT05914935, NCT03896568, NCT02026271, NCT02798406), Prostate Cancer (NCT02555397, NCT01931046, NCT00583024, NCT04097002, NCT00583752, NCT04374240), lung cancer (NCT06618391, NCT02879760), melanoma (NCT04217473, NCT03003676, NCT05664139, NCT05222932) and other cancers (Table 1). The adenovirus vector vaccines that have made relatively rapid progress are A and Recombinant Human Adenovirus (H101). H101 is the world’s first approved virus drug and has an anti-tumor effect on liver cancer. In a Phase 4 clinical trial (NCT05124002) [51], the study aimed to further verify the efficacy and safety of H101 combined with the chemotherapy drug HAIC in the treatment of intrahepatic massive cholangiocarcinoma. Previous studies have demonstrated that the progression-free survival (PFS) of HAIC in the treatment of unresectable intrahepatic cholangiocarcinoma is approximately 8 to 10 months, and the one-year progression-free rate is about 40%. The combined treatment of H101 and HAIC is expected to further enhance the therapeutic effect and increase the PFS.

##### Poxvirus

Poxvirus is a double-stranded DNA virus, which can replicate in cells without entering the nucleus and without the risk of gene integration, greatly improving safety [52]. In addition, poxviruses can also insert large foreign genes (25 KB), thus achieving the expression of complex eukaryotic sequences and multiple genes in mammalian cells, ensuring correct post-translational modifications [52]. Because poxviruses have strong immunogenicity and can mask the immune response to the antigens they carry when used as vaccine vectors, subsequently attenuated poxviruses with modified and deleted virulent genes, have been used as vaccine vectors, such as modified vaccinia virus Ankara (MVA) [53]. In a preclinical study, the two prostate cancer-related antigens mPSCA and mSTEAP1 vaccines carried by MVA demonstrated excellent anti-tumor activity in tumor-bearing mouse models [54]. Moreover, carrying both antigens simultaneously had a stronger inhibitory effect on tumors than carrying either mPSCA or mSTEAP1, which demonstrated the advantage of poxviruses carrying multiple antigens simultaneously [54]. JX-594 (Pexa-Vec) is a vaccine based on the varicella virus. In a Phase 1 clinical trial (NCT00629759), JX-594 demonstrated significantly superior complete remission and systemic efficacy for large-volume tumors compared to other similar drugs [55,56]. Reactions at the injection site of JX-594 were observed in the tumor at all doses. However, systemic tumor responses and delivery to distant tumors through the blood require high doses [56]. In a Phase 2 clinical trial (NCT00554372), researchers explored the efficacy of intratumoral injection of high-dose (10^9^ PFU) and low-dose (10^8^ PFU) JX-594 in patients with liver cancer [57]. The results showed that the median overall survival (OS) in the high-dose group reached 14.1 months compared with 6.7 months in the low-dose group [57]. In terms of safety, JX-594 was generally well tolerated at two doses, and no treatment-related deaths were reported [57]. There is still one study of JX-594 entering Phase 3 clinical trials. However, since the clinical benefit of JX-594 plus sorafenib in the treatment of advanced hepatocellular carcinoma (HCC) did not increase and the effect was worse compared with sorafenib alone, the interim analysis failed to reach the primary endpoint and was terminated early. The combined therapy strategy for oncolytic viruses still needs further exploration [58].

##### Other Virus

In addition to adenovirus vectors and poxvirus vectors, vaccines based on other viral vectors have also been applied to tumors (Table 1). In addition to adenovirus vectors and poxvirus vectors, vaccines based on other viral vectors have also been applied to tumors. For example, lentiviral vector vaccines such as Lenti-HPV-07 have been used in clinical studies to treat HPV-associated oropharyngeal squamous cell cancer [59]. In addition, there are also some vaccines based on other viral vectors, such as Vvax001 (Semliki Forest Virus), HSV G207C (herpes simplex virus-1), etc., which are used in clinical studies to treat cervical intraepithelial neoplasia and brain tumors [60,61].

##### Combination Therapy

Therapeutic cancer vaccines delivered by viral vectors are also inhibited by immunosuppressive factors (such as Treg cells and MDSCs) in the tumor microenvironment, which may weaken the therapeutic effect of the vaccine. To improve efficacy, some research has focused on developing strategies that combine viral vaccines with other therapies to address immunosuppression. In a preclinical study, adenovirus vector-delivered tumor neoantigen vaccine combined with anti-PD-1 antibodies significantly enhanced tumor immunogenic, neoantigen-specific CD8^+^ T-cell response and extended overall survival in MC38 tumor-bearing mice [62]. In addition, adenovirus vector-based tumor vaccines in combination with other therapies have been used in clinical trials to treat melanoma (NCT03003676, NCT05664139, NCT05222932) [63,64,65], colon cancer (NCT04166383, NCT06283134) [66,67], glioblastoma (NCT02798406, NCT02026271) [68,69], lung cancer (NCT06125197, NCT06618391, NCT02879760) [70,71,72], pancreatic cancer (NCT03281382, NCT02894944, NCT02705196) [73,74,75], etc. Clinical studies on combined therapy based on other viral vector vaccines are listed in Table 1. vaccines-13-00672-t001_Table 1Table 1Clinical study of viral vector vaccines updated in recent 5 years.NameCancerROACombination TherapyNCI NumberPhaseRef**Adenovirus Vector-Based Therapeutic Cancer Vaccine**Ad5-yCD/mutTKSR39rep-ADPGlioblastomai.t./NCT056867981[76]Recombinant L-IFN adenovirus injection (YSCH-01)GlioblastomaIntracapsular/NCT059149351[77]DNX-2401GlioblastomaIntra-arterial/NCT038965681[78]Ad-RTS-hIL-12Glioblastomai.t.VeledimexNCT020262711[69]DNX-2401Glioblastomai.t.Anti-PD-1NCT027984062[68]Ad5 peptide transduction domain (PTD)(CgA-E1AmiR122)Neuroendocrine tumorsIntrahepatic artery/NCT027493311/2[79]NG-641, a tumor-selective transgene-expressing adenoviral vectorEpithelial tumorsi.v./NCT040532831[80]NG-350AEpithelial tumori.v./NCT038525111[81]Ad5-yCD/mutTKSR39rep-hIL12Prostate canceri.p./NCT025553971[82]Ad5-SGE-REIC/Dkk3Prostate cancer//NCT019310461[83]Adenovirus/PSA vaccineProstate cancers.c./NCT005830242[84]ORCA-010Prostate canceri.t./NCT040970021/2[85]Adenovirus/PSA vaccineProstate cancers.c.Androgen deprivation therapyNCT005837522[86]AdNRGMProstate canceri.t.CB1954NCT043742401[87]KD01Cervical canceri.t./NCT065525981[88]Human adenovirus 5 injection (d1-d5)Cervical canceri.t.ChemotherapyNCT064550462[89]Adenoviral-mediated interferon-beta (BG00001)Pleural malignanciesi.p./NCT002999621[90]Adenovirus-hIFN-betaPleural malignanciesi.p./NCT000664041[91]Ad5CMV-p53 geneLung cancer//NCT000036491[92]Ad5 (CEA/MUC1/Brachyury)NeoplasmsProstate cancerLung cancerBreast cancerColon cancers.c./NCT033843161[93]Adenovirus (ColoAd1)Colon cancerNon-small-cell lung cancerBladder cancerResectable renal cell carcinomai.t./i.v./NCT020532201[94]GVAXSarcomaRenal cell carcinomamelanoma//NCT002586871[95]Ad/PNPHead and neck canceri.t./NCT037549331/2[96]EnadenotucirevRectal canceri.v.ChemoradiotherapyNCT039165101[97]rAd-IFNPleural mesotheliomai.p.Celecoxib + GemcitabineNCT037108763[98]SCH 721015Mesotheliomai.p.ChemotherapyNCT011196641[99]H101Hepatocellular carcinomai.t.TACENCT058728412[100]H101Hepatocellular carcinomai.t.Tislelizumab and LenvatinibNCT062535982[101]H101Hepatocellular carcinomaHepatic arterial infusion/NCT066853542[102]H101Hepatocellular carcinomai.t.SorafenibNCT051132904[103]HAIC of FOLFOXHepatocellular carcinomaHepatic artery/NCT037800493[104]SynOV1.1Hepatocellular carcinomai.t./NCT046125041[105]VB-111Colorectal canceri.v.Anti-PD-1NCT041663832[66]BioTTT001Colorectal cancer/Anti-PD-1+ RegorafenibNCT062831341[67]BioTTT001Gastric canceri.p.SOX+ Anti-PD-1NCT062831212[106]Recombinant human adenovirus (H101)Cholangiocarcinomai.t.FOLFOXNCT051240024[51]Adenovirus VCN-01RetinoblastomaIntravitreal/NCT03284268Not applicable[107]Ad5/3-E2F-d24-hTNFa-IRES-hIL2 (TILT-123)Ovarian cancer/Anti-PD-1NCT052713181[108]Ad5CMV-p53 geneOvarian canceri.p./NCT000034501[109]Ad5/3-E2F-d24-hTNFa-IRES-hIL2Melanoma//NCT042174731[110]ONCOS-102Melanomai.t.Cyclophosphamide+ Anti-PD-1NCT030036761[63]Recombinant human adenovirus type 5Melanoma/Anti-PD-1+Nab-paclitaxelNCT056641392[64]Ad5/3-E2F-d24-hTNFa-IRES-hIL2MelanomaHead and neck squamous cell carcinoma/Anti-PD-L1NCT052229321[65]Recombinant human adenovirus type 5Lung canceri.t.Chemotherapy + Anti-PD-1NCT066183912[71]Ad-MAGEA3Lung canceri.m.Anti-PD-1NCT028797601/2[72]Ad5/3-E2F-d24-hTNFa-IRES-hIL2 (TILT-123)Lung cancer/Anti-PD-1NCT061251971[70]NG-641Epithelial tumori.v.Anti-PD-1NCT050437141[111]NG-350AEpithelial tumori.v.Anti-PD-1NCT051654331[112]NG-350ARectal canceri.v.ChemoradiotherapyNCT064598691[113]Ad5-yCD/mutTKSR39rep-hIL12Pancreatic canceri.t.ChemotherapyNCT032813821[73]Ad5-yCD/mutTKSR39rep-ADPPancreatic cancer/ChemotherapyNCT028949441[74]Adenovirus serotype 5/35 encoding TMZ-CD40L and 4-1BBL (LOAd703)Pancreatic adenocarcinomaOvarian cancerBiliary carcinomaColorectal canceri.t.ChemotherapyNCT032259891/2[114]LOAd703Pancreatic adenocarcinomaOvarian cancerBiliary carcinomaColorectal canceri.t.Chemotherapy or gemcitabineNCT032259891/2[114]LOAd703Pancreatic canceri.t.Anti-PD-L1NCT027051961[75]Theragene^®^, Ad5-yCD/mutTKSR39rep-ADPPancreas cancer/RadiationNCT047390462[115]Adenoviral p53 (Ad-p53)Solid tumorsi.t.Anti-PD-1/Anti-PD-L1NCT035447232[116]CAdVECSolid tumorsi.t.HER2-specific autologous CAR-T cellsNCT037402561[117]YSCH-01Solid tumorsi.t./NCT051808511[118]Ad5/3-E2F-d24-hTNFa-IRES-hIL2Solid tumors//NCT046953271[119]AdAPT-001Solid tumorsi.t./NCT046739422[120]**Poxvirus Vector-Based Therapeutic Cancer Vaccine**PROSTVAC-V/FProstate cancer/GM-CSFNCT013224903[121,122]PROSTVAC-V/FProstate cancers.c.Anti-PD-1NCT029332551/2[123]TG4050Ovarian carcinomas.c./NCT038395241[124]TG4050Head and neck cancers.c./NCT041831661/2[125]**Other Vector-Based Therapeutic Cancer Vaccine**Lenti-HPV-07HPV-associated oropharyngeal squamous cell cancer, cervical canceri.m./NCT063199631/2[59]Nous-209 genetic vaccineMicrosatellite unstable solid tumors/Anti-PD-1NCT040413101/2[126]Vvax001 therapeutic cancer vaccineCervical intraepithelial neoplasiai.m./NCT060158542[127]HSV G207Recurrent supratentorial brain tumorsi.t./NCT024578451[127]**Abbreviation:** Subcutaneous injection (s.c.); intramuscular injection (i.m.); intravenous injection (i.v.); intertumoral injection (i.t.); intraperitoneal injection (i.p.).


#### 2.1.2. Bacterial Vector Tumor Vaccine

Since bacteria can naturally accumulate on tumors and regulate immune responses, it is believed that bacteria have great potential as carriers for tumor vaccines [128,129,130,131]. Redenti et al. developed a vaccine using the probiotic *Escherichia coli* Nissle 1917 as the tumor neoantigen vector, which significantly enhanced safety and immunogenicity, effectively activated the systemic anti-tumor immune response dominated by T cells, and killed the primary tumor and distant metastases [132]. This system utilizes the properties of living drugs to deliver tumor-specific neoantigens in the optimal environment to induce specific, effective, and long-lasting systemic anti-tumor immunity, such as promoting the activation of dendritic cells, neoantigen-specific T cells, and natural killer cells, as well as significantly reducing tumor-infiltrating immunosuppressive bone marrow cells and regulatory T-cell and B-cell populations [132]. Importantly, vaccines based on bacterial vectors have another advantage in that they can be administered orally. For instance, a preclinical study found that oral administration of the modified Salmonella typhimurium VNP20009 induced a significant anti-cancer effect in B16F10 melmelanoma tumor-bearing mice. Moreover, oral administration has less toxicity and is more reversible compared to intraperitoneal administration. This study indicates that oral administration, as a new approach for bacterial application, has a high degree of safety and efficacy [133].

Nowadays, the bacteria mainly used for preparing tumor vaccines include Salmonella, Listeria, Clostridium, Bifidobacterium, etc. However, many studies are still in the preclinical stage, and few have been translated into clinical practice. ADXS11-001 is an inactivated and attenuated Listeria vector vaccine based on the HPV16 E7 antigen developed by Advaxis. In a Phase 2 clinical study, ADXS11-001 demonstrated good safety and tolerability in patients with cervical cancer [134]. The median overall survival was comparable in the ADXS11-001 group (8.28 months) and the ADXS11-001 + cisplatin group (8.78 months), and the progression-free survival (6.10 months vs. 6.08 months) and the overall response rate (17.1% vs. 14.7%) were also similar [134]. ADXS11-001 was generally well tolerated, and the severity of adverse events was mainly mild to moderate [134]. ADXS11-001 is also being used in a Phase 2 clinical study (NCT02399813) for the treatment of anorectal cancer [135]. Notably, a Phase 3 clinical trial for cervical cancer (NCT02853604) is in a terminated state (for unknown reasons) [136]. There are also some other therapeutic cancer vaccines based on bacterial vectors that have been applied in clinical trials, such as for the treatment of pancreatic cancer (NCT01417000, NCT04589234) [137,138], breast cancer (NCT06631092) [139], and other solid tumors (Table 2).

### 2.2. Cellular Vaccines

#### 2.2.1. Dendritic Cell Vaccine

Dendritic cells are specialized antigen-presenting cells (APCs) that initiate effective tumor-specific immune responses by phagocytosis and processing of tumor antigens to T cells [147,148,149,150]. DC vaccine is obtained by sensitizing DC cells through tumor cell DNA, RNA, tumor cell lysate, tumor antigen protein/polypeptide, and other substances, and then using the powerful presentation function of DC cells to activate the patient’s T-cell immune response to achieve the purpose of tumor control [151]. At present, most DC vaccine products use patients’ autologous peripheral blood monocytes, which are prepared through in vitro expansion and antigen loading [152]. DC-based vaccines have been widely selected for immunotherapy. Currently, four DC vaccine products have been approved worldwide, including Hybricell (Genoa Biotechnologia), CreaVaxPCC (CreaGene), DCVax-Brain (Northwest Biotherp), and APCEDEN (APAC Biotech) for the treatment of melanoma, prostate cancer, kidney cancer, and glioma. In addition, based on the international clinical trial register platform (http://www.clinicaltrials.gov), according to the data shows that many based on DC vaccines have entered clinical trials, as a clinical trial has entered the stage 3 (NCT00045968), shows a good application prospect [153]. Most of the rest are Phase 1–2 clinical studies (Table 3). DC’s vaccines are mainly used in clinical trials to treat liver cancer, lung cancer (NCT02688673, NCT05195619) [154,155], breast cancer (NCT02063724, NCT02061423, NCT06435351, NCT04879888, NCT04105582) [156,157,158,159,160], melanoma (NCT01622933, NCT02301611, NCT01808820, NCT02678741, NCT01876212) [161,162,163,164,165], hematological malignancies (NCT02528682) [166], ovarian carcinoma (NCT05714306) [167], lung cancer (NCT02956551, NCT04147078, NCT03871205, NCT03371485) [168,169,170,171], glioblastoma (NCT03914768, NCT02771301, NCT04888611, NCT02529072, NCT02366728) [172,173,174,175,176], gastric cancer (NCT04567069, NCT04147078) [169,177], hepatocellular carcinoma (NCT04147078) [169], colorectal cancer (NCT04147078, NCT06545630, NCT03730948, NCT01885702) [169,178,179,180], and so on.

However, the clinical efficacy of DC vaccines is very limited, and recently, efforts have been made to develop new strategies to enhance the efficacy of DC vaccines. DC vaccine is developing towards individuation and precision, combination with other therapies, and integration with new technologies. In personalized and precise treatment, tumor-specific neoantigens with high immunogenicity can be predicted and screened according to the genetic information of patients’ tumor tissues so as to customize DC vaccines that are more in line with patients’ own characteristics, improve efficacy, and reduce side effects. In 2015, the first personalized neoantigen DC vaccine was tested in Phase 1 clinical trials (NCT00683670) [181]. They selected seven neoantigens from melanoma patients, loaded them into DC isolated from PBMC, and injected them intravenously three times to enhance the T-cell immune response. All three patients treated survived, and no adverse reactions were observed, demonstrating the safety and feasibility of the personalized neoantigen DC vaccine. Another personalized neoantigen DC vaccine trial was conducted in patients with advanced non-small-cell lung cancer (NCT02956551) [182]. Similarly, loading patients’ personalized neoantigens into DC isolated from the PBMC showed an overall 25% objective response rate and 75% disease control rate, with only mild and transient side effects observed. In addition, there are several other neoantigen DC vaccines for the treatment of ovarian cancer [183], breast cancer (NCT04879888, NCT04105582) [159,160], lung cancer (NCT04078269, NCT02956551, NCT03871205, NCT03205930) [168,170,184,185], liver cancer (NCT03674073) [186], and so on. As technology continues to advance, DC vaccines will focus more on individualized and precise strategies, with the deepening of research on the combined application of DC vaccines with immune checkpoint inhibitors, chemotherapy, and radiotherapy. Combination therapy will become the main trend of DC vaccine development. For example, a trial showed that the pp65 pulse DC vaccine combined with the chemotherapy drug temozolomide for glioma significantly extended overall survival (41.1 months) [187]. In another trial, an autologous EPHA2-targeted CAR-DC vaccine loaded with TP53 mutant peptide (TP53-EPHA-2-CAR-DC) combined with an anti-PD-1 antibody/anti-CTLA4 antibody is used in patients with locally advanced/metastatic solid tumors or relapsed/refractory lymphoma (NCT05631886) [188]. DC vaccine combined with immune checkpoint inhibitors can enhance the immune response of T cells. When combined with chemotherapy, more tumor antigens are released by the killing effect of chemotherapy drugs on tumor cells, and the DC vaccine can reactivate immune cells and improve the clearance effect of tumor cells. Based on the advantages of combination therapy, the synergies of DC vaccine and more therapies will continue to be explored and optimized to form better treatment options to overcome the limitations of tumor efficacy. In addition, with the development of nanotechnology, gene editing technology, cell engineering technology, etc., DC vaccines are also deeply integrated into these new technologies. For example, Mao et al. successfully delivered Cas9 mRNA and sgRNA to DC cells using LNP, achieving effective gene editing on DC cells [189]. By gene editing, the PD-L1 of DC cells was effectively knocked out, the activation and maturation of DC cells were enhanced, and the anti-tumor immune response mediated by T cells was improved, which significantly inhibited the growth of colon cancer in the tumor-bearing mouse model [189]. Another study showed that DC vaccines loaded with CircRNA encoding tumor antigens (FAPα and survivin) induced a stronger CD8^+^ T-cell response [190]. Moreover, its combination with gemcitabine significantly inhibited Panc02 tumor growth (89% inhibition rate) and extended survival in mice [190]. A more efficient antigen delivery vector based on nanotechnology was developed to improve the efficiency of antigen uptake and presentation by DC cells. And DC cells were modified by gene editing technology to enhance their immune activation ability.

Although DC vaccines show great potential in cancer immunotherapy, there are still challenges in preparation techniques, individual differences, off-target effects, delivery efficiency, and immunosuppressive microenvironments. However, with the advancement of technology, the continuous development of new cell separation and preparation technology, gene editing technology, efficient delivery systems, etc., will make the DC vaccine expected to become an important breakthrough in cancer immunotherapy. vaccines-13-00672-t003_Table 3Table 3Clinical study of DC-based vaccines updated in recent 5 years.NameCancerROACombination TherapyNCI NumberPhaseRefAutologous dendritic cells pulsed with tumor lysate antigenGlioblastomai.d./NCT000459683[153]Autologous AdHER2-transduced dendritic cell vaccineBreast canceri.d./NCT017301181[191]Placental or tumor-derived heat shock protein gp96-induced DCsSolid tumorss.c.i.t./NCT064776141[192]Autologous EphA2-targeting CAR-DC vaccine loaded with KRAS mutant peptideSolid tumorsi.v.AbraxaneCyclophosphamideAnti-PD-1Anti-CTLA4NCT056318991[193]Autologous EphA2-targeting CAR-DC vaccine loaded with TP53 mutant peptideSolid tumorsLymphomasi.v.AbraxaneCyclophosphamideAnti-PD-1Anti-CTLA4NCT056318861[188]Immune-modified DCMultiple myeloma Plasmacytoma//NCT064359101[194]Tumor antigen-pulsed DCEsophageal squamous cell carcinomas.c./NCT053173251[195]DC loaded with autologous tumor homogenateGlioblastomai.d.TemozolomideNCT045236882[196]Autologous genetic-modification-free DC cells will be loaded with multiple tumor neoantigen peptidesGlioblastomas.c./NCT062532341[197]Tumor antigen-sensitized DCMelanomaBladder cancerColorectal cancers.c./NCT052356071[198]Tumor neoantigen peptide vaccine/neoantigen-based DCAdvanced malignant solid tumorss.c./NCT05749627Not applicable[199]Autologous DC loaded with patient-specific peptides or tumor lysatesOvarian carcinoma/CyclophosphamideNCT057143061/2[167]Dendritic cell with tumor-associated antigen and patient-specific neoantigensOvarian cancer//NCT052707201[200]Tumor antigen-sensitized DC vaccineColorectal cancers.c./NCT065456301[178]DC vaccines loaded with HPV 16/18 E6/E7 epitopesCervical intraepithelial neoplasia//NCT038701131[201]Anti-HER2/HER3 dendritic cell vaccineBreast canceri.d.Anti-PD-1NCT043487472[202]Autologous dendritic cell-adenovirus p53 vaccineBreast cancers.c./NCT000826411/2[203]Total tumor RNA-pulsed DCsMedulloblastomai.d.Td vaccineautologous HSCsAnti-PD-1NCT065148981[204]Immune-modified dendritic cells fused with leukemic cells (DCvac)B-cell acute lymphoblastic leukemia//NCT052626731[205]Autologous dendritic cellProstate cancers.c./NCT055332031[206]Immune-modified dendritic cell vaccine (DCvac)T-cell acute lymphoblastic leukemia//NCT052777531[207]Peptide-pulsed autologous dendritic cellBreast canceri.d./NCT061956181[208]HER2-pulsed dendritic cell vaccineHER2-positive breast canceri.d.Anti-her2Anti-PD-1T-cell therapyNCT053784641[209]Dendritic cell vaccine loaded with circular RNA encoding cryptic peptideHER2-negative advanced breast canceri.d.Anti-PD-1NCT065300821[210]MIDRIX4-lung autologous DC vaccineNon-small-cell lung canceri.v.Antigen-specific DTHNCT040821821[211]Autologous dendritic cell (ADC) vaccineSmall-cell lung canceri.d.CarboplatinADC vaccineNCT044877561/2[212]TTRNA-DC vaccines with GM-CSFMedulloblastomai.d.Td vaccineautologous HSCsAnti-PD-1NCT065148981[204]Tumor lysate-loaded autologous DC vaccineColorectal canceri.d./NCT065229192[213]Autologous dendritic cell vaccine loaded with personalized peptides (PEP)Pancreatic adenocarcinomas.c./NCT046272461[214]HER-2-pulsed DC1HER2-positive breast cancer/Anti-HER2Anti-PD-1PaclitaxelNCT053256322[215]Allogeneic dendritic cell vaccine (DCP-001)Ovarian cancer//NCT047395271[216]Autologous DC loaded with autologous tumor homogenateMesotheliomai.d.Anti-PD-1Interleukin-2NCT035464261[217]HER2 targeting autologous dendritic cell (AdHER2DC) vaccineEndometrial canceri.d.Anti-PD-1N-803LenvatinibNCT062534941/2[218]Autologous dendritic cell (DC) vaccineLiver canceri.m.Anti-PD-L1Anti-VEGFRTPneumococcal vaccineNCT039423281/2[219]Multiple signals-loaded dendritic cells vaccineHepatocellular carcinomai.v.CyclophosphamideNCT043172482[220]Autologous DCs pulsed with mutated peptidesColorectal canceri.v./NCT037309481[179]Autologous tumor blood vessel antigen (TBVA)-dendritic cell vaccineKidney canceri.d.CabozantinibNCT051278242[221]Autologous DCs pulsed with genetically modified tumor cells or tumor-related antigens including neoantigensGlioblastomai.d./NCT039147681[172]CCL21Non-small-cell lung canceri.m.Anti-PD-1NCT035463611[222]HER2-sensitized DCBreast canceri.d./NCT036308092[223]DC/multiple myeloma (MM) Fusion vaccineMultiple myeloma/Anti-PD-1NCT037820642[224]PDC*lung01Non-small-cell lung cancers.c.i.v.Anti-PD-1Antifolate agentsNCT039707461/2[225]MG-7 antigenGastric cancers.c.Anti-PD-1NCT045670691/2[177]Autologous tumor lysate-pulsed dendritic cell vaccinationGlioblastomai.d.Anti-PD-1Poly-ICLCNCT042018731[226]Tumor antigen-sensitized DC vaccineEsophagus cancers.c./NCT050239281[227]DC loaded with tri-antigens (WT1/TERT/survivin)Acute myeloid leukemia//NCT05000801Not applicable[228]DCs pulsed with GSC antigens (GSC-DCV)Recurrent glioblastomas.c.Anti-PD-1NCT048886112[174]DC vaccine loaded with personalized peptidesNon-small-cell lung cancers.c.CyclophosphamideNCT051956191[155]Neoantigen-loaded DCLung cancers.c./NCT063299081[229]Autologous DCs loaded with multiple tumor neoantigen peptidesGlioblastoma multiforme of braini.dTemozolomideNCT049683661[230]NeoantigenHepatocellular carcinomaColorectal canceri.d.Anti-PD-1NCT049127652[231]Neoantigen-derived dendritic cellRefractory Tumors.c.Anti-PD-1LenvatinibNCT057676841[232]Neoantigen-primed DCGastric cancerHepatocellular carcinomaNon-small-cell lung cancerColon rectal cancers.c./NCT041470781[169]Neoantigen-loaded DCNon-small-cell lung cancers.c./NCT038712051[170]Neoantigen dendritic cellBreast cancerInguinal or axillaryLeukapheresisNCT064353511[158]Tumor neoantigen-based vaccine FRAME-001Non-small-cell lung cancers.c./NCT049984742[233]Neoantigen-pulsed dendritic cellBreast cancer//NCT041055821[160]Autologous neoantigen-targeted dendritic cellNon-small-cell lung canceri.v.Antigen-specific DTHNCT040782691[184]Peptide-pulsed dendritic cellBreast canceri.d./NCT048798881[159]Neoantigen-pulsed dendritic cellBreast cancer//NCT041055821[160]Personalized DC vaccineGastric cancerHepatocellular carcinomaNon-small-cell lung cancerColon rectal cancers.c./NCT041470781[169]Neoantigen-loaded DC vaccineColorectal cancer//NCT018857021/2[180]**Abbreviation:** Intradermal injection (i.d.); subcutaneous injection (s.c.); intramuscular injection (i.m.); intravenous injection (i.v.).


#### 2.2.2. Tumor Cell Vaccine

Based on the characteristics of tumor cells carrying all tumor antigen information, the use of tumor cells as vaccines can provide adequate antigen information to the patient’s immune system, eliminating the need to identify the optimal antigen in a specific type of cancer, overcoming the problem of tumor antigen loss, and thus helping to better activate the anti-tumor immune response [234]. The types of tumor cell-based vaccines mainly include autologous tumor cell vaccines and allogeneic tumor cell vaccines.

##### Autologous Tumor Cell Vaccine

Autologous tumor cell vaccines belong to the category of personalized tumor therapeutic vaccines, which are mainly tumor cells obtained from patients, and the tumorigenic ability of tumor cells is removed by irradiation while retaining their immune activity. The treated tumor cells contain tumor-associated antigens, which can activate the patient’s own immune system after being transfused into the patient, prompting the body to produce a specific immune response against tumor cells and achieve the purpose of tumor treatment. It is worth noting that vaccines prepared by directly inactivating tumor cells have poor immunogenicity and very limited efficacy. To address the problem, current strategies are to genetically modify tumor cells and combine them with adjuvants or other therapies to improve the anti-tumor efficacy of vaccines. For example, Chang et al. developed a tumor cell vaccine that overexpresses mesothelin (a new tumor antigen for ovarian cancer), which, in combination with IL-12, significantly increased the proportion of mesothelin-specific T cells and prolonged mouse survival [235]. Currently, more research is on autologous tumor cell vaccines expressing GM-CSF (GVAX). In a variety of mouse tumor models, GVAX has been shown to promote the antigen presentation and activation of DC and has a good curative effect [236,237,238]. GVAX has been used in clinical trials for the treatment of pancreatic cancer (NCT02243371, NCT03153410, NCT00389610) [239,240,241], prostate cancer (NCT00140374) [242], and other tumors (Table 4). In addition, GVAX has also been selected for use in combination with other therapies to improve efficacy in clinical trials. For example, combination with nivolumab and ipilimumab for neuroblastoma (NCT04239040) [243], combination with Cyclophosphamide for Pancreatic Cancer (NCT01417000) [137], and combination with Pembrolizumab for Colorectal Cancer (NCT02981524) [244], and so on (Table 4). In Table 2, we systematically list the updated clinical studies of autologous tumor cell-based vaccines in the past five years.

##### Allogeneic Tumor Cell Vaccines

Allogeneic whole tumor cell vaccines usually contain two or three established human tumor cell lines to overcome the limitations of antigen source, molecular expression, and standardization of production and preparation of autologous tumor cell vaccines [245]. For allogeneic tumor cell vaccines, batch preparation of tumor cell lines or allogeneic cells can be achieved, and their cost is much lower than that of individualized vaccines. Moreover, allogeneic tumor cell vaccines usually carry multiple tumor-associated antigens, increasing the probability of covering more patients. For tumors with low immunogenicity, the immunogenicity of vaccines can be enhanced through genetic modification to demonstrate better therapeutic effects. Like VACCIMEL, a therapeutic cancer vaccine approved in Argentina composed of four allogeneic melanoma cell lines, effectively induces T-cell immune responses against neoantigens, allogeneic antigens, and tumor-associated antigens [246]. In a Phase 2 clinical study (NCT01729663), VACCIMEL demonstrated significant benefits in distant metastasis-free survival (DMFS) in patients with cutaneous melanoma receiving adjuvant therapy [247,248]. VACCIMEL combined with Bacillus Calmette–Guerin (BCG) and recombinant human granulocyte macrophage-colony stimulating factor (rhGM-CSF) adjuvants induced a strong specific immune response to TAA in patients and significantly enhanced the therapeutic effect of the vaccine [247,248,249,250]. Few allogeneic tumor cell therapeutic cancer vaccines have entered clinical research and are basically in the 1–2 stage, mainly used for the treatment of glioblastoma (NCT03360708, NCT04642937, NCT06305910, NCT04388033) [251,252,253,254].vaccines-13-00672-t004_Table 4Table 4Clinical study of tumor cells-based vaccines updated in recent 5 years.TargetCancerROACombination Therapy**NCI Number****Phase****Ref****Autologous tumor cellular vaccine**GM-CSF-secreting autologous neuroblastoma cell vaccine (GVAX)Neuroblastoma/Anti-PD-1Anti-CTLA4NCT042390401[243]GVAX pancreas vaccinePancreatic canceri.d.Anti-PD-1CRS-207NCT022433712[239]GVAX pancreas vaccinePancreatic canceri.d.Anti-PD-1IMC-CS4NCT031534101[240]GVAX pancreas vaccinePancreatic canceri.d./NCT003896102[241]GVAX pancreas vaccinePancreatic cancer/Anti-PD-1NCT031613792[255]GVAX pancreas vaccinePancreatic cancer/Anti-PD-1Anti-CTL4NCT031902652[256]GVAX pancreas vaccinePancreatic canceri.d.Cyclophosphamide FOLFIRINOXNCT015953212[257]GVAX pancreas vaccinePancreatic cancer/CyclophosphamideAnti-PD-1NCT026482822[258]GM-CSF-secreting autologous leukemia cell vaccination (GVAX)Myelodysplastic syndromeAcute myeloid leukemiaChronic myelomonocytic leukemiai.d.ChemotherapyNCT017733952[259]GM-CSF-secreting leukemia cell vaccinationsMyeloid leukemias.c. or i.d./NCT00426205Not applicable[260]Allogeneic myeloma GM-CSF vaccineMultiple myelomai.d.LenalidomidePneumococcal vaccineNCT033764772[261]GVAX colon vaccineColorectal canceri.d.Anti-PD-1CYNCT029815242[244]Allogeneic colon cancer cell vaccine (GVAX)Colorectal canceri.d.CYSGI-110NCT019662891[262]Colon GVAXColorectal cancer/CYNCT006561231[263]Particle-delivered, allogeneic tumor cell lysate vaccine (PalloV-CC)Colorectal canceri.d./NCT038279671
GVAX prostate cancer vaccineProstate canceri.d.CYNCT016968771/2[264]Autologous tumor cellular vaccineProstate canceri.d.
NCT066366822
GVAXMelanomaSarcoma/renal cell carcinoma//NCT002586871[95]Personalized neoantigen cancer vaccineKidney cancers.c.
NCT029507661
Autologous breast cancer cells engineered to secrete GM-CSFBreast cancer//NCT003176031[265]Autologous breast cancer cells engineered to secrete GM-CSFBreast cancer//NCT008804641[266]GRT-C901, GRT-R902Non-small-cell lung cancerColorectal cancer Gastroesophageal adenocarcinomaUrothelial carcinoma/Anti-PD-1Anti-CTL4NCT036397141/2[267]GRT-C901, GRT-R902Non-small-cell lung cancerColorectal cancer Gastroesophageal adenocarcinomaUrothelial carcinoma/Anti-PD-1Anti-CTL4NCT036397141/2[267]OVM-200Prostate cancerLung cancerOvarian cancer//NCT051045151[268]**Allogeneic tumor cell vaccine**Therapeutic vaccine (ACIT-1)Pancreatic cancerOther cancer//NCT030960931/2[269]Malignant glioma tumor lysate-pulsedGlioblastomas.c.Autologous dendritic cellNCT033607081[251]Allogeneic tumor lysate vaccine (GBM6-AD)Glioblastoma/CD200AR-LimiquimodNCT046429371[252]Allogeneic tumor lysate vaccine (GBM6-AD)Glioblastoma/CD200AR-LimiquimodNCT063059101[253]DC/tumor cell fusion vaccineGlioblastoma/Anti-CTLA4NCT043880331/2[254]Therapeutic vaccine (ACIT-1)Pancreatic cancerOther cancer//NCT030960931/2[269]**Abbreviation:** Intradermal injection (i.d.); subcutaneous injection (s.c.).


### 2.3. Peptide Vaccines

Peptide tumor vaccine uses synthetic peptide fragments as antigens to stimulate the body to produce an anti-tumor immune response. Peptide vaccines have been paid more and more attention to because they are completely synthetic, with high safety (no complete pathogen), high specificity, flexible design, and low cost [151]. Currently, three peptide vaccines are marketed worldwide, vitespen, EGF-P64K, and racotumomab, for the treatment of glioma, renal cell carcinoma, cervical cancer, and non-small-cell lung cancer. Furthermore, many peptide tumor vaccines are in the clinical trial stage. We summarize the clinical research progress of the updated peptide tumor vaccines in the past five years in Table 5.

Traditional peptide tumor vaccine has some defects, such as poor immunogenicity, low efficacy, and short half-life, which affect its therapeutic effect in clinical application. To address the very limited efficacy of peptide tumor vaccines, many studies have focused on screening highly specific neoantigen peptides, optimizing immune-stimulating adjuvants, developing more effective delivery systems, and exploring combination therapy strategies to enhance immune response and tumor suppression.

Personalized neoantigen vaccines have been regarded as an effective method for inducing, enhancing, and diversifying anti-tumor T-cell responses [270]. For example, a personalized neoantigen polypeptide vaccine demonstrated clinical feasibility, safety, and immunogenicity for the first time in a Phase I clinical trial in melanoma patients [271]. The vaccine can target up to 20 predicted individual tumor neoantigens, increasing the number of antigen-specific T cells, such as induced CD4^+^ and CD8^+^ T cells targeting 58 (60%) and 15 (16%) of 97 unique neoantigens, respectively [271]. It is well known that there is still no better treatment method for patients with glioblastoma. After standard treatment, there are often problems of recurrence, poor treatment effect, and limited survival period. In a study, through somatic mutation analysis of the tumors of 173 glioblastoma patients, personalized peptide vaccines targeting tumor-specific neoantigens were produced [272]. Among the blood samples of 97 (90%) monitored patients, vaccine-induced immune responses to at least one vaccination peptide were detected in 87 cases [272]. Most patients developed persistent specific T-cell responses, and the survival period (53 months) of patients with multiple vaccine-induced T-cell responses was significantly longer than that of patients with no or low induced responses (27 months) [272]. This study demonstrated the feasibility of individualized neoantigen-targeted peptide vaccines, which provide promising potential treatment options for the treatment of glioblastoma patients [272]. With advances in high-throughput sequencing technology, genomics, synthesis technology, and data science, rapid screening, optimization, and preparation of personalized antigens can be achieved. Based on the optimization of tumor neoantigen personalized vaccine design strategy, many related types of vaccines have been used in clinical trials to treat melanoma (NCT05098210, NCT01970358, NCT03929029), lung cancer (NCT04397926, NCT02897765, NCT04487093, NCT03380871), and other cancers (Table 5).

GM-CSF is a powerful immune adjuvant that can increase the maturation and function of dendritic cells, thereby enhancing antigen presentation [273]. In a preclinical study, local injection of GM-CSF, IL-2, and HPV16 E7 peptide enhanced vaccine-specific immune responses and induced higher CTL and cytokine release without increasing immunosuppressive Treg cells, more effectively inhibiting the growth of TC-1 tumor cells [274]. In a clinical study (Phase 2, NCT02636582), a peptide vaccine composed of HER2-derived MHC Class I peptide E75 (nelipepimut-S, NPS) combined with GM-CSF adjuvant in the treatment of patients with ductal carcinoma in situ (DCIS) showed good vaccine tolerance and relatively good safety [275]. Moreover, vaccination enhances the NPS-specific cytotoxic T lymphocyte (CTL) response, and the increase in the proportion of specific T cells produced in the NPS + GM-CSF group exceeds that in the NPS alone treatment group [275]. Cytosine-guanosine oligodeoxynucleotide (CpG) also is a strong adjuvant that promotes the production of pro-inflammatory cytokines, stimulates DC and B-cell activation, and induces and enhances Th1 type immune response [276,277,278,279,280,281,282,283]. In a study, all eight melanoma patients with HLA-A2^+^ showed rapid and intense antigen-specific T-cell responses after receiving treatment with a low-dose CpG 7909 combined with melanoma antigen A analog peptide and incomplete Freund’s adjuvant vaccine [284]. The number of antigen-specific T cells produced by patients in the CPG treatment group was significantly higher than that in the CpG treatment group [284]. The mechanism is achieved by the increased T cells recognizing and killing melanoma cells in an antigen-specific manner [284]. Other different antigen-peptide vaccines combined with adjuvants have also been used in clinical trials to treat melanoma (NCT00471471, NCT00112242, NCT00112229, NCT05098210) [284,285,286,287,288,289], breast cancer (NCT02593227, NCT05232916, NCT03012100, NCT05098210) [289,290,291,292], lung cancer (NCT02818426, NCT03380871, NCT01949701, NCT06472245) [293,294,295,296], glioma (NCT02193347) [297], pancreatic cancer (NCT03645148, NCT05013216) [291,298], and other cancers (Table 5).

Although neoantigen peptide vaccines have great potential in tumor immunotherapy, their progress in clinical trials has been hindered due to the limitations of antigen cell uptake and cross-presentation. Based on the development of delivery technology, nanovaccines co-delivered with neoantigens and adjuvants have been regarded as a very promising approach to personalized cancer immunotherapy, with encouraging results in several preclinical animal models [299,300,301,302,303]. For example, Moon et al. designed a high-density lipoprotein-mimicking nanodiscs delivery strategy that co-delivered neo-epitopes and the adjuvant CPG, significantly improved the delivery efficiency of antigen in vivo, improving delivery efficiency and enhancing the frequency of neoantigen-specific CD8α+ cytotoxic T lymphocytes (47 times higher), and effectively inhibiting the tumor growth of B16F10 and MC38 tumor-bearing mice [300]. In addition, some nanovaccines based on co-delivery antigens and adjuvants have also been used to treat melanoma [302,304], breast cancer, colon cancer [302,303,305,306], liver cancer [307], lung cancer [308], gliomas [309], etc. However, neoantigen and adjuvant tumor vaccines loaded based on new delivery technologies are still mainly preclinical studies.

In addition to strategies such as optimizing adjuvants and developing new delivery systems to enable peptide tumor vaccines, combination with other therapies is also an important approach. In a Phase 2 clinical trial (NCT02455557), the peptide vaccine SurVaxM plus temozolomide in glioblastoma patients showed a good safety profile, a strong antigen-specific CD8^+^ T cells response, and 95.2% of patients remained progression-free six months after diagnosis [310]. Glioblastoma is a very-high-mortality tumor, and in clinical trials evaluating standard radiation and chemotherapy, the median survival of most patients was only 14.6 to 16.0 months. It is exciting to see that SurVaxM plus temozolomide treatment significantly improved the median overall survival of patients (25.9 months) [310,311,312]. For patients with metastatic melanoma, improving overall survival has been a formidable challenge to overcome. In a Phase 3 clinical trial (NCT00094653), the median overall survival of patients with metastatic melanoma treated with glycoprotein 100 (gp100) peptide vaccine alone was 6.4 months. To improve survival, the combination of the gp100 peptide vaccine and ipilimumab (an anti-CTLA-4 antibody) showed good clinical expectations, extending survival to 10.0 months [313]. In another Phase 1b clinical study (NCT02897765), NEO-PV-01, a neoantigen vaccine tailored to a patient’s tumor gene mutation, was shown to be effective in combination with PD-1 antibodies in patients with advanced melanoma, non-small-cell lung cancer, and bladder cancer [314]. In addition, some other clinical studies related to peptide tumor vaccines combined with other therapies in recent years are summarized in Table 5.

At present, the research progress of peptide tumor vaccines mainly revolves around the research of personalized peptide vaccines, tumor-associated antigens, and adjuvants (such as TLR agonists, STING agonists, cytokines) and delivery systems (such as nanoparticles, liposomes, and other novel delivery systems) to enhance immune response. With the development of sequencing technology and bioinformatics, new adjuvants, new delivery systems, and other technologies, the trend of personalized and combination therapy of peptide vaccines is developing. However, peptide tumor vaccines also face many challenges, such as poor immunogenicity, tumor immunosuppressive microenvironment, individual differences, and antigen escape. It is believed that with the innovation and development of technology, peptide tumor vaccines will definitely achieve accurate vaccine design by combining multiple omics and exploring multi-mode combined treatment schemes to improve the clinical effect of vaccines.vaccines-13-00672-t005_Table 5Table 5Clinical study of peptide tumor vaccines updated in recent 5 years.Target AntigenAdjuvantCancerRoACombination TherapyNCI NumberPhaseRefGP96 heat shock protein–peptide complex/Liver cancer//NCT042062542/3[315]Tumor antigen peptides/Liver cancers.c./NCT050598211[316]ELI-002 7P/Solid tumorss.c./NCT057268641/2[317]ELI-002 2P (Amph modified KRAS peptides, Amph-G12D and Amph-G12R admixed with admixed Amph-CpG-7909)/Kirsten rat sarcoma (KRAS) mutated pancreatic ductal adenocarcinoma and other solid tumorss.c./NCT048530171[318]Neoantigen peptides vaccine/Non-small-cell lung cancers.c./NCT043979261[319]ARG1 peptidesMontanide ISA-51Solid tumorss.c./NCT036891921[320]HLA-A*2402 or A*0201 restricted peptidesMontanide ISA 51Solid tumorss.c./NCT019496881/2[321]HLA-A*0201restricted URLC10 peptidesMontanide ISA 51Non-small-cell lung cancers.c./NCT019497011/2[295]Two peptides called UCP2 and UCP4 derived from telomeraseMontanide ISA 51Non-small-cell lung cancer//NCT028184261/2[293]OSE2101Montanide ISA 51Non-small-cell lung cancers.c./NCT064722453[296]Melan-A-ELA + NY-ESO-1b + MAGE-A10 peptide + Montanide + CpGMontanide ISA 51Melanoma//NCT001122421[287]PD-L1 peptideMontanide ISA 51Multiple myelomas.c./NCT030427931[322]IDH1 peptide vaccineGM-CSFGliomai.d./NCT021933471[297]FRα peptideGM-CSFBreast canceri.d./NCT025932272[290]HER2/neu peptide GLSI-100 (GP2 + GM-CSF)GM-CSFBreast canceri.d./NCT052329163[291]Multi-epitope folate receptor alpha peptideGM-CSFBreast canceri.d./NCT030121002[292]Neoantigen peptidesGM-CSFSolid tumors//NCT036628151[323]Neoantigen peptidesGM-CSFPancreatic cancer//NCT036451481[324]Mutant Kirsten rat sarcoma (KRAS)-targeted long peptidePoly-ICLCPancreatic cancer//NCT050132161[298]NEO-PV-01 (personalized neoantigen)Poly-ICLCMelanomaNon-small-cell lung cancers.c./NCT028977651[314,325]Neoantigen peptidesPoly-ICLCBreast cancerMelanomai.m./NCT050982101[289]Neoantigen peptidesPoly-ICLCMelanoma//NCT019703581[326]AE37 peptide vaccine/Breast canceri.d.Anti-PD-1NCT040248002[327]OTSGC-A24/Gastric cancers.c.Anti-PD-1 +Anti-CTLA4NCT037840401[328]Synthetic tumor-associated peptide/Pancreatic cancerColorectal cancers.c.Anti-PD-1Anti-PD-1 + APX005MNCT026009491[329]Neoantigen peptide/Non-small-cell lung cancers.c.EGFR-TKIAnti-angiogenicNCT044870931[330]Liposomal HPV-16 E6/E7 multi-peptide vaccine PDS0101/HPV-oropharyngeal squamous cell carcinomas.c.Anti-PD-1NCT052328511/2[331]Neoantigen heat shock protein vaccine (rHSC-DIPGVax)/Glioma/Anti-PD-1 +Anti-CTLA4NCT049438481[332]Survivin long peptide (SurVaxM)Montanide ISA 51Neuroendocrine tumorss.c.Octreotide acetateNCT038796941[333]UCP2 and UCP4 derived from telomerase (UCPVax)Montanide ISA 51Papillomavirus-positive cancerss.c.Anti-PD-L1NCT039463582[334]NPMW-peptide vaccineMontanide ISA 51Myelodysplastic syndromeAcute myeloid leukemia/Anti-PD-L1NCT027509951[335]Personalized multi-peptide vaccine cocktailsXS15, Montanide ISA 51Cancers.c.TLR1/2 ligand XS15NCT05014607
[336]MVF-HER-2 (597–626) and MVF-HER-2 (266–296)Montanide ISA 720Advanced solid tumorsi.m./NCT064147331[337]Neoantigen peptides vaccineMontanide ISA 51 + Poly-ICLCMelanoma/Anti-PD-1+Anti-CTLA4NCT039290291[338]PVX-410 (contains four synthetic peptides)Poly- ICLCSmoldering multiple myelomas.c.Citarinostat + LenalidomideNCT028860651[339]NEO-PV-01Poly-ICLCNon-small-cell lung cancers.c.Anti-PD-1 +ChemotherapyNCT033808711[294]Pooled mutant KRAS-targeted long peptide vaccinePoly-ICLCColorectal cancerPancreatic cancer/Anti-PD-1 +Anti-CTLA4NCT041170871[340]DNAJB1-PRKACA fusion kinase peptidePoly-ICLCLiver cancer/Anti-PD-1 +Anti-CTLA4NCT042485691[341]Personalized multi-peptidePoly-ICLCProstate cancer/CDX-301NCT050102001[342]KRAS peptide vaccinePoly-ICLCNon-small-cell lung cancer/Anti-PD-1+Anti-CTLA4NCT052541841[343]MUC1 peptide vaccinePoly-ICLCDuctal carcinoma in situs.c.Aromatase inhibitorNCT062183031[344]Galinpepimut-SGM-CSFAcute myelogenous leukemiaOvarian cancerColorectal cancerBreast cancer Small-cell lung cancer/Anti-PD-1NCT037619141/2[345]Neoantigen peptideGM-CFSSolid tumorsi.v.Anti-PD-1NCT052693811/2[346]**Abbreviation:** Intradermal injection (i.d.); subcutaneous injection (s.c.); intramuscular injection (i.m.); intravenous injection (i.v.).


### 2.4. Nucleic Acid Vaccines

#### 2.4.1. DNA Tumor Vaccine

In cancer therapy, DNA cancer vaccines are considered to be a very attractive and promising means, with advantages such as low cost, cell-independent production, durable immune response, and potential to target multiple neoantigens [151,347]. Of course, there are also defects of host gene integration risk, autoimmune reaction risk, and low transfection efficiency [151]. In order to improve efficacy and safety, different strategies are being used to optimize and improve DNA vaccines. To improve efficacy and safety, efforts have been made to optimize and improve DNA vaccines through different strategies, such as inserting optimized optimal antigens.

Previous studies have shown that selecting and inserting the optimal antigen for plasmid DNA is an ideal way to enhance vaccine immunogenicity and induce a broad immune response, which can overcome problems associated with antigen loss, modification, and tolerance [347]. DNA vaccine construction based on enhanced immunogenicity strategy mainly includes chimeric DNA vaccine, neoantigen DNA vaccine, and polypeptide DNA vaccine. Chimeric DNA vaccines are heterologous antigenic vaccines that encode proteins or peptides from different species, and their sequences have significant homology with the self-ortholog [348,349]. Since the homologous and natural protein sequences are only similar but not identical, this helps to circumvent immune tolerance while maintaining homology that can be recognized by T cells to enhance the potential immunogenic response [348,349,350]. Previous studies have shown that xenoantigens are more effective than autoantigens [350,351]. For example, xenogeneic DNA vaccines targeting human tyrosinase were approved to treat canine melanoma [349], Xenovaccines designed with rhesus CEA (rhCEA) as the immunogen against human carcinoembryonic antigen (hCEACAM-5 or commonly hCEA) can activate CD4^+^ T cells and autoreactive CD8^+^ T cells, and produce high-titer antibodies against hCEA and have significant anti-tumor effects. Furthermore, codon-optimized RhCEA cDNA (rhCEAopt) was demonstrated to have higher immune reactivity than hCEAopt in mice [352], Chimeric rat/human HER2 efficiently circumvents HER2 tolerance in cancer patients [353]. DNA vaccines encoding mouse/human chimeric proteins induce a better immune response against Erbb-2 tumors in mice [354]. DNA xenovaccines have shown encouraging results in a clinical trial for melanoma [355,356]. Neoantigen vaccines are selected to express antigens specifically in tumor tissue, which overcomes the problem of immune tolerance deficiencies and side effects [357,358]. For example, Li et al.’s optimized polypeptide neoantigen DNA vaccine induced strong neoantigen-specific T-cell responses in preclinical mouse breast cancer models E0771 and 4T1 and combined with anti-PD-L1 antibody effectively inhibited the growth of E0771 tumors and maintained anti-tumor immunity [359].

In clinical trials, DNA vaccines are being used to treat liver cancer (NCT04251117) [360], melanoma (NCT03655756) [361], breast cancer (NCT05455658, NCT04246671, NCT02780401) [362,363,364], non-melanoma skin cancers (NCT04160065) [365], glioblastoma (NCT04015700, NCT05743595) [366,367], prostate cancer (NCT03532217, NCT03600350, NCT04090528) [368,369,370], and other cancers (Table 6), most of which were in the Phase 1–2 clinical research stage. Despite efforts to improve the delivery efficiency of DNA vaccines, their immunogenicity in clinical trials remains limited. Therefore, people still need to continue exploring more strategies to enhance the immunogenicity of DNA vaccines, such as optimizing DNA vaccine vectors, combining cytokine adjuvants, and exploring innovative delivery methods, etc. [371].

#### 2.4.2. RNA Vaccine

With the outbreak of COVID-19, the urgent use of two mRNA vaccines has brought mRNA vaccines back into the spotlight. Like DNA, mRNA can encode an unlimited number of proteins and peptides. However, mRNA vaccines have several irreplaceable advantages, such as no risk of gene integration, repeatability, coding flexibility and versatility, short production cycle, and low cost [383,384,385]. Based on the editable flexibility of mRNA vaccines, they can encode tumor antigens as tumor antigen vaccines, cytokines for immunotherapy, tumor suppressors to inhibit tumor development, chimeric antigen receptors for engineered T-cell therapy, and genomic proteins for gene therapy. In this section, we will focus on describing the progress of mRNA therapeutic cancer vaccines in clinical studies (Table 7).

Because mRNA is easily degraded by RNases, there is little research on naked mRNA vaccines, and the main focus is on the application of delivery systems to deliver mRNA into the body. Currently, the strategies for delivering mRNA mainly include protamine, cationic liposomes, and LNP. Protamin-coated mRNA vaccines use the positive charge of protamine to form a complex with negatively charged mRNA to avoid mRNA degradation [386]. For example, in a Phase 1/2 clinical trial (NCT00204607), subcutaneous injection of protamine-stabilized mRNAs encoding Melan-A, Tyrosinase, gp100, Mage-A1, Mage-A3, and survivin in 21 patients with metastatic melanoma demonstrated that the vaccine was safe with no grade II adverse events and activated the immune response. The frequency of Foxp3^+^/CD4^+^ immunosuppressive cells was significantly decreased, and some patient-specific T cells were increased [387]. The strategy of delivering mRNA into the body by means of an mRNA-lipoplex complex formed by cationic liposomes with negatively charged mRNA is currently studied and paid more attention. For example, BNT-111, developed by BioNtech Company, is a mRNA-lipoplex vaccine designed for melanoma antigen (MAGE-A3, NY-ESO-1, TPTE, Tyrosinase). In a Phase II clinical study (NCT02410733), BNT-111 demonstrated good clinical benefits, with 75% of patients producing an anti-tumor immune response [388]. Lipid nanoparticles are currently very mature mRNA delivery platforms, mainly composed of lipids, phospholipids, and cholesterol [389,390]. mRNA-4157, developed by Moderna, is an mRNA vaccine encoding 34 tumor neoantigens and wrapped with LNP. It is also the fastest-growing mRNA therapeutic cancer vaccine (Phase 3, NCT06077760, NCT05933577) [391,392]. In a 2b clinical trial (NCT03897881), the recurrence-free survival of melanoma patients treated with mRNA-4157 combined with pembrolizumab was longer than that of pembrolizumab monotherapy (79% versus 62%). And it has relatively good safety, with no mRNA-4157-related grade 4/5 events [393]. Furthermore, another Phase 1 clinical study (NCT03313778) on non-small-cell lung cancer or melanoma evaluated the safety, tolerability, and immunogenicity of mRNA-4157 [394]. The results showed that no patient had grade 4/5 adverse events or dose-limiting toxicity [394]. mRNA-4157 alone can induce consistent new generation and enhance the pre-existing T-cell response to targeted neoantigens, and the combination therapy induces sustained neoantigen-specific T-cell responses and the expansion of cytotoxic CD8 and CD4 T cells [394]. The relevant clinical studies of mRNA-4157 have demonstrated the great potential and significance of mRNA-4157 as an adjuvant monotherapy or in combination with other therapies.

There are also many other mRNA therapeutic cancer vaccines in the clinical stage, which are used to treat melanoma (NCT04526899, NCT03897881) [395,396], liver cancer (NCT05981066, NCT05738447, NCT05761717) [397,398,399], lung cancer (NCT03164772, NCT06735508) [400,401], pancreatic cancer (NCT06326736, NCT06577532, NCT06496373, NCT06156267, NCT06353646, NCT04161755) [402,403,404,405,406,407], and other cancers (Table 7). In addition, many studies are exploring the design and application of novel mRNA, such as self-amplified mRNA (saRNA), trans-amplified mRNA (taRNA), and circular mRNA (circRNA), as well as the long-term preservation means of mRNA nanoparticles, drug delivery routes, and organ-selective precision translation [383]. These explorations are expected to enable mRNA-based anti-cancer therapies to further cover various types of cancer and benefit a broad population of patients. vaccines-13-00672-t007_Table 7Table 7Clinical study of mRNA vaccines updated in recent 5 years.NameCancerROACombination TherapyNCI NumberPhaseRefNY-ESO-1, MAGE-A3, tyrosinase, and TPTEMelanomai.v.Anti-PD-1NCT045268992[395]mRNA-4157Melanoma/Anti-PD-1NCT038978812[396]mRNA-4157Melanomai.m.Anti-PD-1NCT059335773[392]mRNA-4157Cutaneous squamous cell carcinomai.m.Anti-PD-1NCT062958092/3[408]mRNA-4157Renal cell carcinomai.m.Anti-PD-1NCT063074312[409]HBV mRNA vaccineLiver canceri.m./NCT057384471[398]Neoantigen mRNA vaccine (ABOR2014/IPM511)Liver canceri.m./NCT05981066Not applicable[397]Neoantigen mRNA personalized cancer vaccineLiver cancers.c.Anti-PD-1NCT05761717Not applicable[399]mRNA-4157Non-small-cell lung canceri.m.Anti-PD-1NCT060777603[391]BI 1361849 mRNA vaccine comprises 6 drug product components (MUC1, survivin, NY-ESO-1, 5T4, MAGE-C2, MAGE-C1)Non-small-cell lung canceri.d.Anti-PD-L1Anti-CTLA4NCT031647721/2[400]BI 1361849 mRNA vaccine comprises 6 drug product componentsNon-small-cell lung canceri.d.Anti-PD-L1Anti-CTLA4NCT031647721/2[400]Neoantigen mRNA vaccinesNon-small-cell lung cancer/Anti-PD-L1NCT067355081[401]Fixed combination of shared cancer antigensHead and neck canceri.v.Anti-PD-L1NCT045342052[410]EBV mRNA vaccineMalignant tumorsi.m./NCT057147481[411]Personalized neoantigen mRNA vaccine iNeo-Vac-R01Digestive system neoplasmss.c./NCT060197021[412]mRNA neoantigen vaccine iNeo-Vac-R01Digestive system neoplasmss.c./NCT060267741[413]Neoantigen mRNA vaccinesDigestive system neoplasmss.c./NCT03468244Not applicable[414]Neoantigen mRNA vaccines iNeo-Vac-R01Neoantigen mRNA vacciness.c./NCT060268001[415]Neoantigen mRNAEsophageal cancerNon-small-celllung cancers.c./NCT03908671Not applicable[416]mRNA neoantigen vaccine (mRNA-0523-L001)Endocrine tumori.m./NCT06141369Not applicable[417]Neoantigen mRNA vaccinesPancreatic cancer/Gemcitabine + AbraxaneNCT063267361[402]KRAS neoantigen mRNA vaccine (ABO2102)Pancreatic canceri.m.Anti-PD-1NCT065775321[403]Neoantigen mRNA vaccinesPancreatic cancer/Anti-PD-1NCT064963731[404]Neoantigen mRNA vaccinesPancreatic cancer/Anti-PD-L1NCT061562671[405]XH001 (neoantigen cancer vaccine)Pancreatic cancer/Anti-CTLA4 + ChemotherapyNCT06353646Not applicable[406]Personalized neoantigen tumor vaccinesPancreatic cancer/Anti-PD-L1NCT041617551[407]mRNA 2752CarcinomaIntralesional (IL)Anti-PD-1NCT028720251[418]mRNA-4157Solid tumorsi.m.Anti-PD-1NCT033137781[419]Neoantigen mRNA vaccineSolid tumorsi.t./NCT061953841[420]Neoantigen mRNA vaccine SW1115C3Solid tumorss.c./NCT051987521[421]Neoantigen mRNA personalized cancer vaccineSolid tumorss.c.Anti-PD-1NCT05949775Not applicable[422]Neoantigen mRNA vaccinesSolid tumorsi.m.Anti-PD-1NCT064970101[423]XH001 (neoantigen cancer vaccine)Solid tumors/Anti-PD-1NCT05940181Not applicable[424]Individualized neoantigen vaccine mRNA-4157Solid tumorsi.m.Anti-PD-1NCT033137781[419]IL-7, IL-12 BNT152 + 153Solid tumorsi.v./NCT047100431[425]mRNA-2752, a lipid nanoparticle encapsulating mRNAs encoding human OX40L, IL-23, and IL-36γSolid tumorsi.m.Anti-PD-1NCT037399311[426]IL-12 MEDI1191Solid tumorsi.t./NCT039468001[427]**Abbreviation:** Intradermal injection (i.d.); subcutaneous injection (s.c.); intramuscular injection (i.m.); intravenous injection (i.v.); intertumoral injection (i.t.).


## 3. Challenges and Trends in Therapeutic Vaccines

Immunotherapy is an effective means of treatment following drug, surgery, and radiotherapy, and its clinical role is increasingly prominent. Therapeutic cancer vaccines, as one of the main methods of immunotherapy, have become a new growth point of biomedicine with broad industrial prospects in the post-COVID-19 era. Major international vaccine companies [such as BioNTech SE, CureVac AG, Moderna TX, Merck Sharp & Dohme Corp] have laid out research and development pipelines to promote their clinical transformation.

For immunotherapy strategies, the anti-tumor process mainly consists of three links: effective antigen release, immune activation, and tumor killing. These links complement each other, and none can be missing. Vaccines developed in the past often had many deficiencies, resulting in slow development and limited therapeutic effects. Tumor antigens are the key factors that initiate the anti-tumor immune response and also the crucial link that tumor therapeutic vaccines need to address. For immune checkpoint inhibitors (PD-1/PD-L1 antibodies) and CAR-T cell therapy, they address the aspect of “tumor killing”, and the treatment process faces the problem of immune tolerance. Recently, the development of gene sequencing technology and bioinformatics has enabled more precise identification of specific gene mutations and neoantigens in patients’ tumor cells, thereby promoting the increasing precision of antigens. Therapeutic cancer vaccines will be highly customized based on the individual tumor antigen characteristics of each patient to enhance the vaccine’s specificity and efficacy. However, individualized vaccines based on tumor neoantigens still face many challenges, such as tumor heterogeneity, immunogenicity, and how to scientifically and reasonably design and validate clinical trial protocols, etc. Tumor cells are highly heterogeneous, so the antigen expression of tumor cells in different patients may vary. In practical applications, it is difficult to find a universal tumor antigen for vaccine design, which also increases the difficulty for vaccines to cover all tumor cells. Furthermore, tumors progress rapidly and are prone to mutation, which requires a fast process from antigen sequencing and screening to design, undoubtedly putting pressure on vaccine production. The issue of immunogenicity is that tumor antigens usually have weak immunogenicity and are difficult to stimulate a strong immune response, and the immune system in the body may develop immune tolerance to tumor antigens, resulting in poor vaccine efficacy. For instance, a personalized neoantigen cancer vaccine based on mRNA was terminated due to its clinical efficacy failing to meet expectations (Phase 1/2, NCT03480152) [428]. Therefore, in order to solve the problems in the immune activation part, many studies have been dedicated to developing more effective adjuvants and antigen presentation techniques to enhance the immunogenicity of tumor antigens and break immune tolerance. In particular, peptide vaccines are greatly affected by adjuvants. Many studies have proved the favorable effects of adjuvants on vaccines, such as adjuvants GM-CSF, CpG, etc. Moreover, adjuvants are also developing towards compound adjuvants, taking advantage of their respective strengths and complementing each other’s weaknesses. It is believed that with the advantages of new adjuvants and compound adjuvants, the efficacy of vaccines is expected to be continuously improved in the future. In addition to adjuvants, new delivery technologies have also been a key focus area in recent years. So far, delivery carriers include viruses, bacteria, cells, lnp, etc. For viral vector vaccines, the development of genetic engineering technology has made the modification of viral vectors safer, more precise, and more efficient, which can improve the targeting, immunogenicity, and safety of the vectors. For instance, by designing and optimizing the structure and function of viruses through gene editing, viral vectors can infect tumor cells more specifically while reducing their impact on normal cells. There are already many therapeutic cancer vaccines based on viral vectors in the clinical research stage (Table 1). However, such vaccines still face many challenges, mainly the accompanying issues related to immunogenicity and safety. For example, the safety issues brought about by potential inserted gene mutations, the neutralizing antibodies produced by antiviral responses reduce the therapeutic effect and the production complexity problems such as the high cost of large-scale production and quality control of viral vectors. Delivery technologies such as cell and LNP are all aimed at improving the delivery efficiency of antigens into tumors, increasing the effective concentration of antigens within the tumor to enhance the activation of immune responses, and simultaneously reducing off-target effects outside the tumor to improve safety. In addition to improving delivery technology, combination therapy is also a mainstream trend in overcoming cancer. Whether based on cells or LNP or other carriers, therapeutic cancer vaccines combined with immune checkpoint inhibitors, chemotherapy, radiotherapy, etc., have demonstrated significant advantages in preclinical and clinical studies to exert a synergistic effect and improve therapeutic outcomes.

Nowadays, research is focused on technological breakthroughs in various therapeutic cancer vaccines. By integrating the characteristics of multiple technologies and the continuously accumulated clinical experience, therapeutic cancer vaccine therapy has great potential and application space in the field of cancer treatment. However, future research still requires further improvement and optimization in aspects such as antigen screening, vector design, and production and preparation processes. Strive to reduce production costs, enhance the accuracy of antigens, improve the efficiency and targeting of delivery systems, and verify their long-term efficacy and good safety through large-scale clinical trials.

Therapeutic cancer vaccines, particularly highly personalized neoantigen vaccines, represent a pivotal advancement in cancer immunotherapy. However, they also encounter significant economic and regulatory hurdles that must be overcome to achieve widespread clinical adoption and long-term sustainability. Economically, the primary challenge for personalized vaccines lies in their bespoke nature. The production of autologous vaccines necessitates tumor sequencing for each individual patient, predictive screening of neoantigens, personalized vaccine design, and manufacturing. This intricate process involves compliance with Good Manufacturing Practice (GMP) standards, costly single-lot production, extended cycle times that may delay treatment initiation, and stringent quality-control protocols, all contributing to prohibitively high unit costs (potentially reaching hundreds of thousands or even millions of dollars per patient). Even with allogeneic or shared neoantigen vaccines, which can reduce costs through economies of scale, challenges persist in ensuring universal efficacy across diverse patient populations and minimizing “off-target” toxicity. High development expenses, encompassing complex clinical trials and substantial infrastructure investments, such as decentralized manufacturing facilities or cold-chain logistics networks, ultimately translate into elevated treatment costs, posing critical challenges to patient accessibility and the financial sustainability of healthcare systems.

In the regulatory dimension, the traditional drug approval paradigm is mainly based on uniform, replicable product characteristics and phase III trial data from large-scale homogeneous populations. However, the essence of personalized vaccines—that each patient’s product is different—poses a fundamental challenge to the current regulatory framework. Regulatory agencies (such as the FDA, EMA, etc.) need to address a key issue: defining the “consistent” quality, safety, and efficacy of products based on “individual customization rather than standardization”. This requires an innovative transformation of the regulatory paradigm: developing alternative endpoints based on biomarkers or immune responses to accelerate the approval path; building a new real-world evidence (RWE) collection and evaluation system to accumulate efficacy evidence using individualized data; formulating CMC (Chemistry, Manufacturing, and Controls) guidelines for personalized therapeutic products, emphasizing process control and platform validation rather than the physical consistency of individual products; exploring adaptive licensing pathways, allowing for more flexible initial approval and subsequent data confirmation within a strict framework. Additionally, mechanisms for generating post-market evidence and data sharing after accelerated approval are also crucial.

## 4. Conclusions and Future Directions

Over the past few decades, with the continuous breakthroughs in immunology and precision medicine technologies, people’s understanding of how cancer cells evade immune system monitoring and their roles in the body has been greatly enhanced. This has led to significant progress in tumor immunotherapy, which is constantly developing in a favorable direction for defeating cancer. Previous immune checkpoint inhibitors and cell therapy methods have demonstrated their ability to regress tumors in some studies on hematological malignancies and solid tumors. These advancements have shown the feasibility of applying tumor immunotherapy and therapeutic cancer vaccines. With the continuous development of technology, humans will be able to more accurately identify highly immunogenic neoantigens in the future. Especially the personalized neoantigen vaccines that are currently regarded as having great potential, such vaccines have the advantages of unassailable high specificity and strong immune activation ability. Personalized neoantigen vaccines can avoid attacking normal tissues and significantly reduce off-target toxicity by targeting mutations specific to tumor cells (neoantigens) [271]. In a study, the feasibility, safety, and immunogenicity of a vaccine targeting up to 20 predicted personalized tumor neoantigens were demonstrated, and the vaccine-induced multifunctional CD4^+^ and CD8^+^ T cells targeted 60% (58) and 16% (15) of the 97 unique neoantigens, respectively [271]. This indicates that personalized vaccines have achieved the distinction between mutant antigens and wild-type antigens. Furthermore, neoantigens have not been cleared by the central immune tolerance mechanism, so they are more likely to activate the T-cell response, which is conducive to exerting anti-tumor effects. Personalized neoantigen vaccines are designed based on the gene mutation maps of cancer patients. They activate the CD8^+^ and CD4^+^ T cells of the patient’s own immune system by synthesizing mRNA or DNA to encode tumor-specific neoantigens, thereby achieving precise attacks on tumor cells. Take the mRNA-4157 vaccine as an example. Its fastest-growing personalized mRNA vaccine is delivered to the body through the LNP delivery platform to induce a strong T-cell immune response. In a Phase 2b trial, the combination therapy of mRNA-4157 and Keytruda reduced the risk of recurrence or death in patients with stage III/IV melanoma by 44% and enhanced T-cell-mediated tumor cell destruction [429]. To date, many vaccine platforms targeting personalized neoantigens have entered the clinical trial stage, mainly including vaccines based on peptides, DNA, RNA, adenoviruses, and DC cells. It is worth noting that these vaccines have triggered T-cell immune responses against cancer-related targets, but they face the challenge of low overall immunogenicity. It is believed that with the continuous improvement of various technologies and the development of new ones, combined with targeted and efficient delivery technologies, highly personalized and universal therapeutic vaccines can be developed for different situations. Moreover, through combined treatment, a synergistic effect can be achieved to maximize the therapeutic effect.

## Figures and Tables

**Figure 1 vaccines-13-00672-f001:**
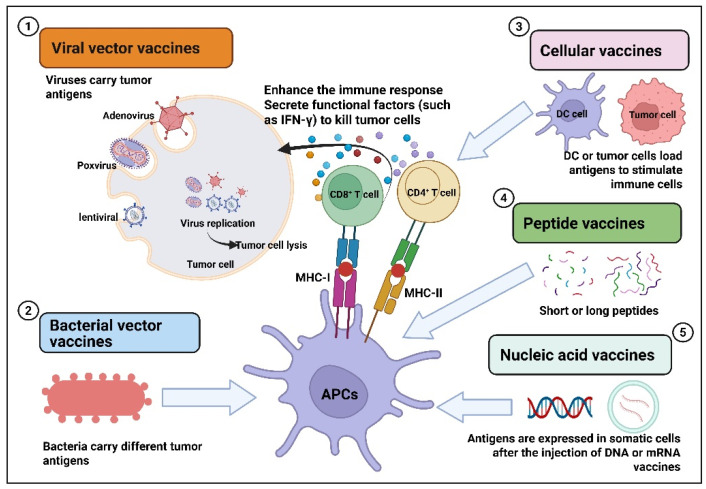
Mechanism of action of therapeutic cancer vaccines.

**Table 2 vaccines-13-00672-t002:** Clinical study of bacterial vector vaccines updated in recent 5 years.

Name	Cancer	ROA	Combination Therapy	NCI Number	Phase	Ref
NECVAX-NEO1	Solid tumors	Orally	Anti-PD-1/PD-L1	NCT06631079	1/2	[140]
NECVAX-NEO1	Triple-negative Breast cancer	Orally	Anti-PD-1 nab-paclitaxel chemotherapy	NCT06631092	1/2	[139]
ADXS11-001	Cervical cancer	i.v.	/	NCT01266460	2	[141]
ADXS11-001	Cervical cancer	i.v.	/	NCT02164461	1	[142]
ADXS11-001	Anal cancer Rectal cancer	i.v.	/	NCT02399813	2	[135]
CRS-207	Pancreatic cancer	i.v.	GVAX vaccine cyclophosphamide	NCT01417000	2	[137]
Saltikva	Pancreatic cancer	Orally	/	NCT04589234	2	[138]
Clostridium Novyi-NT	Solid tumors	i.v.	/	NCT01924689	1	[143]
Clostridium Novyi-NT	Solid tumors	i.v.	Anti-PD-1	NCT03435952	1	[144]
TXSVN	Multiple myeloma	Orally	/	NCT03762291	1	[145]
SGN1	Solid tumors	i.t.	/	NCT05038150	1/2	[146]

**Abbreviation:** Intravenous injection (i.v.); intertumoral injection (i.t.).

**Table 6 vaccines-13-00672-t006:** Clinical study of DNA vaccines updated in recent 5 years.

Target	Cancer	ROA	Combination Therapy	NCI Number	Phase	Ref
Emm55 streptococcal antigen	Melanoma	i.t.	/	NCT03655756	1	[361]
TAEK-VAC-HerBy	Chordoma Breast cancer	i.v.	Anti-HER2	NCT04246671	1/2	[363]
pNGVL4aCRTE6E7L2 DNA vaccine	Cervical neoplasia	i.m.	/	NCT04131413	1	[372]
HPV	Cervical cancer Vulvar cancer Vaginal cancer	/	/	NCT02653118	Observational	[373]
HPV	Cervical cancer	/	/	NCT04588402	Observational	[374]
IGFBP-2, HER2, and IGF1R	Breast cancer	i.d.		NCT02780401	1	[364]
Neoantigen DNA vaccine	Prostate cancer	i.m.	Anti-PD-1 or Anti-CTLA4 + PROSTVAC	NCT03532217	1	[368]
Neoantigen DNA vaccine (GNOS-PV02)	Hepatocellular carcinoma	i.d.	Anti-PD-1	NCT04251117	1/2	[360]
Neoantigen DNA vaccine	Recurrent brain tumor	i.m.	/	NCT03988283	1	[375]
pAc/emm55 (pDNA)	Non-melanoma skin cancers	Intralesionally	/	NCT04160065	1	[365]
Prostatic acid phosphatase (pTVG-HP)	Prostate cancer	i.d.	Anti-PD-1	NCT03600350	2	[369]
pTVG-HP DNA vaccine	Prostate cancer	i.d.	Anti-PD-1	NCT04090528	2	[370]
DNA-PEI polyplex	Neuroblastoma	i.m.	/	NCT04049864	1	[376]
Personalized neoantigen DNA vaccine	Glioblastoma	/	/	NCT04015700	1	[366]
Personalized neoantigen DNA vaccine	Glioblastoma	i.m.	/	NCT05743595	1	[367]
TriAd vaccine	Head and neck cancer	i.v.	Anti-PD-L1/TGF-beta Trap (M7824)	NCT04247282	1/2	[377]
GX-188E HPV DNA vaccine	Head and neck cancer	i.m.	Anti-PD-1	NCT05286060	2	[378]
pING-hHER3FL	Advanced cancer	i.m.		NCT03832855	1	[379]
Neoantigen DNA vaccine	Small-cell lung cancer	i.m.	Anti-PD-L1	NCT04397003	2	[380]
CD105/Yb-1/SOX2/CDH3/MDM2-polyepitope plasmid DNA vaccine	Non-small-cell lung cancer	i.d.	/	NCT05242965	2	[381]
CD105/Yb-1/SOX2/CDH3/MDM2-polyepitope plasmid DNA vaccine	Breast cancer	i.v.	/	NCT05455658	2	[362]
Glypican3 (GPC3)-targeted DNA plasmid vaccine (NWRD06)	Hepatocellular carcinoma	i.m.	/	NCT06088459	1	[382]

**Abbreviation:** Intradermal injection (i.d.); intramuscular injection (i.m.); intravenous injection (i.v.); intertumoral injection (i.t.).

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
