# Peer review of "Research and Clinical Progress of Therapeutic Tumor Vaccines"

_vaccines, 2025, doi:10.3390/vaccines13070672_

Round 1
Reviewer 1 Report
Comments and Suggestions for Authors
Dear Drs. Dong, Dr. Li, and distinguished colleagues:
I have completed the review of your manuscript "Research and Clinical Progress of Therapeutic Tumor Vaccines" (Manuscript ID: vaccines-3323601), submitted to the journal Vaccines. I appreciate the opportunity to evaluate this comprehensive review, which systematically analyzes the current landscape of therapeutic cancer vaccines.
Your manuscript offers a thorough analysis of diverse vaccine platforms, including viral vectors, bacterial vectors, cellular vaccines, peptide vaccines, and nucleic acid vaccines. The extensive compilation of clinical trial data from the past five years represents a significant contribution to the field. The work demonstrates your team's deep knowledge of cancer immunotherapy and provides valuable information for researchers and clinicians.
Strengths of the Manuscript:
The systematic organization and comprehensive coverage of the different vaccine modalities are exemplary. The inclusion of detailed tables documenting recent clinical trials (Tables 1-7) will serve as an invaluable reference for the research community. The analysis of combination therapy strategies, particularly integration with immune checkpoint inhibitors, is timely and relevant. The manuscript is well-written and appropriate for the journal's readership.
Areas for Improvement:
While the manuscript is comprehensive, I believe it would benefit from the following improvements:
- Critical Analysis: The review would be strengthened by a more critical analysis of clinical trial failures. Understanding why certain approaches have not been successful is as important as documenting successes.
- Discussion of Biomarkers: The manuscript lacks an adequate discussion of biomarkers for patient selection and response prediction, which are crucial for personalized cancer vaccine approaches.
- Economic and Regulatory Perspectives: Given the complexity of therapeutic cancer vaccines, particularly personalized approaches, addressing economic feasibility and regulatory pathways would increase the practical value of your review.
- Mechanistic Comparisons: Some sections would benefit from more in-depth mechanistic explanations. A comparative figure illustrating the mechanisms of the different vaccine platforms would enhance the reader's understanding.
- Specific Outcomes: When available, including primary outcomes in your clinical trial tables would add value.
Specific Technical Comments:
- Lines 24-40: The summary could be more specific about the key findings and conclusions.
- Lines 110-200: Consider adding a comparative analysis of the advantages and limitations of viral vectors.
- Line 289: The formatting of Table 3 requires improvements for clarity.
- Lines 600-610: The challenges section needs a more detailed analysis of specific shortcomings.
- Line 567: The conclusion should include more specific recommendations for future research priorities.
Minor Issues:
- Line 45: "In recent decades" should be more precisely defined.
- Line 156: Missing reference to the statement about the CD vaccine.
- Line 445: The section on the mRNA vaccine should address storage/stability issues.
- Minor grammatical corrections are needed.
Sincerely,
Reviewer 2 Report
Comments and Suggestions for Authors
Dear Authors,
Thank you for your insightful and timely review on therapeutic cancer vaccines. I believe the paper can be accepted in current format. However, the following comments and suggestions are provided to help improve the clarity, completeness, and impact of the paper:
- In the introduction, the authors mention the division of tumor vaccines into therapeutic and preventive types. It would strengthen the review if you briefly mention some real-world examples of preventive tumor vaccines that are in clinical use. This will help clarify the distinction for the reader and contextualize the broader scope of cancer vaccines.
-
Please correct the typographical error in line 603:
“mrna” should be rewritten as “mRNA”
-
The section on viral vector vaccines would benefit from a concise explanation of their mechanism of action. For example, you could describe how viral vectors deliver tumor-associated antigens into host cells to elicit cytotoxic T cell responses, and how vectors like adenovirus or vaccinia virus have been utilized.
-
To enhance the forward-looking value of this review, I strongly recommend adding a dedicated “Future Directions” section that explores cutting-edge areas such as AI-driven neoantigen discovery to accelerate and optimize personalized vaccine development, pan-cancer or pan-vaccine strategies targeting shared tumor antigens across multiple malignancies, and a discussion of the evolving paradigm between personalized versus universal cancer vaccines, highlighting both opportunities and challenges associated with each approach.
5. An illustrative figure could be added to provide a visual summary of the study, clearly presenting the classification of therapeutic cancer vaccine types (peptide, mRNA, dendritic cell, viral vector), and their mechanisms of action (immune activation, T cell stimulation, tumor cell lysis).
Reviewer 3 Report
Comments and Suggestions for Authors
Dear Authors,
The article “Research and clinical progress of therapeutic tumor vaccines” introduces the analysis of clinical trials, mainly obtained from the www.clinicaltrials.gov website. It should be noted that the U.S. government does not review or approve the safety and science of all studies listed on this website.
Since the review contains a lot of tables with the references on clinical trials, it is desirable to change them for the real published results if possible.
- It is needed to add at least one figure (for example, the mechanism of action of cancer vaccines or classification of vaccines, advantages, targeted and efficient delivery technologies, and limitations ).
- Please, add some own conclusions: what are the most promising therapeutic cancer vaccine and the reasons for this. Which vaccines are FDA-approved and already widely used in clinical practice since the Abstract and Conclusion sections do not contain concrete information.
- Please, highlight the novelty of the review. There are many reviews on this topic, for example, Ni, L. Advances in mRNA-Based Cancer Vaccines. Vaccines 2023, 11, 1599. https://doi.org/10.3390/vaccines11101599
- The column “Phase” contains the phrase “Not Applicable”. Please, clarify the sense of it.
- Tables 1-3 and 4-7 differ in the size. Please, bring them to the same format
Round 2
Reviewer 3 Report
Comments and Suggestions for Authors
Dear Authors.
The article has been improved.
If the main focus of the article is still on the progress of clinical trials, please, explain, what are the feathures of this progress - the amount of clinical trials or the quality of clinical trials or a new trends (combined therapy, for example) in clinical trials?
It is desirable to disclose the new trends in clinical trials of therapeutic tumor vaccines and include this information to the Abstract section.
The relevant table format issues have NOT been modified in this revision.
Author Response
|
Response 3: Thank you sincerely again for the reviewers' questions. We will explain the questions you raised below. Question 1. If the main focus of the article is still on the progress of clinical trials, please, explain, what are the feathures of this progress - the amount of clinical trials or the quality of clinical trials or a new trends (combined therapy, for example) in clinical trials? Question 2. It is desirable to disclose the new trends in clinical trials of therapeutic tumor vaccines and include this information to the Abstract section. Response 1 and 2: Thank you for your professional advice. The main purpose of our writing this article is to systematically introduce the research progress of various therapeutic tumor vaccines in clinical trials, such as which vaccines have been updated to enter clinical trials in the past five years, what stage they are at in clinical trials, and which vaccines have been combined with other therapies in the clinical stage, etc. These are summarized in the article and tables. However, the suggestions put forward by the reviewers are very important. In this revision, we have improved the relevant content of the abstract section, lines 17-26 on page 1. Furthermore, in our parts 3 and 4, the content of personalized vaccines and combination therapy was highlighted, and in the sections of various vaccines, we also described the research on combination therapy. If the reviewers have any questions or suggestions, they can raise them and we will make further revisions. Thank you. Abstract section: lines 17-26 on page 1: “Various therapeutic cancer vaccines, such as viral vector vaccines, bacterial vector vaccines, cell vector vaccines, peptide vaccines and nucleic acid vaccines, have all been applied in clinical re-search. With the continuous development of technology, therapeutic cancer vaccines are evolving towards the trends of precise antigens, efficient carriers, diversified adjuvants and combined ap-plications. For instance, the rapidly advancing mRNA-4157 vaccine is a typical representative that combines personalized antigens with efficient delivery vectors (lipid nanoparticles, LNP), and it also shows synergistic advantages in melanoma patients treated in combination with immune checkpoint inhibitors. In this article, we will systematically discuss the current research and de-velopment status and clinical research progress of various therapeutic cancer vaccines.”
1. Question 3. The relevant table format issues have NOT been modified in this revision. Response: Thanks to the reviewer’s carefully and thoroughly suggestion. In this revision, we have unified the formats of all the tables. Taking Table 1 and Table 4 as examples, it is as follows:
|

Round 3
Reviewer 3 Report
Comments and Suggestions for Authors
Dear Authors,
The article has been sufficiently improved and can be published.